# Design of modular autoproteolytic gene switches responsive to anti-coronavirus drug candidates

Nik Franko[1], Ana Palma Teixeira[1], Shuai Xue[1], Ghislaine Charpin-El Hamri[2] & Martin Fussenegger [1,3✉]

The main (Mpro) and papain-like (PLpro) proteases encoded by SARS-CoV-2 are essential to process viral polyproteins into functional units, thus representing key targets for anti-viral drug development. There is a need for an efficient inhibitor screening system that can identify drug candidates in a cellular context. Here we describe modular, tunable autoproteolytic gene switches (TAGS) relying on synthetic transcription factors that self-inactivate, unless in the presence of coronavirus protease inhibitors, consequently activating transgene expression. TAGS rapidly report the impact of drug candidates on Mpro and PLpro activities with a high signal-to-noise response and a sensitivity matching concentration ranges inhibiting viral replication. The modularity of the TAGS enabled the study of other *Coronaviridae* proteases, characterization of mutations and multiplexing of gene switches in human cells. Mice implanted with Mpro or PLpro TAGS-engineered cells enabled analysis of the activity and bioavailability of protease inhibitors in vivo in a virus-free setting.

[1] ETH Zurich, Department of Biosystems Science and Engineering, Mattenstrasse 26, CH-4058 Basel, Switzerland. [2] Département Génie Biologique, Institut Universitaire de Technologie, Université Claude Bernard Lyon 1, F-69622 Villeurbanne, Cedex, France. [3] University of Basel, Faculty of Life Science, Basel, Switzerland. ✉email: fussenegger@bsse.ethz.ch

A newly identified member of the *Coronaviridae* family, severe acute respiratory syndrome coronavirus 2 (SARS-CoV-2)[1], is the causative agent of COVID-19[2], which has been responsible for one of the most devastating medical crisis of our time. Due to the lack of effective treatment options, the number of deaths related to COVID-19 has exceeded 3 million as of May 2021. Although the approved vaccines offer highly efficient prophylactic protection against the onset of the disease[3,4], the availability of therapeutic drugs that can be used to treat acutely infected patients is very limited. A substantial number of drugs has been proposed[5], but only remdesivir, which is a polymerase inhibitor initially developed for Ebola infection, has been approved so far[6]. Nevertheless, its efficacy to treat COVID-19 patients remains questionable[7]. Therefore, there is an urgent need to develop compounds directed against viral targets with high specificity and minimal side effects.

Once SARS-CoV-2 viral particles enter the cell, the viral genome in the form of positive-stranded RNA is translated into two polyproteins, pp1a and pp1ab, which are proteolytically cleaved into functional non-structural proteins (nsp) by two virally encoded proteases. The 3C-like protease (3CLpro), also called the main protease (Mpro), cleaves eleven sites in the polyproteins and the papain-like protease (PLpro) processes three cleavage sites. PLpro is also known to play a role against the host immune response to facilitate virus replication[8]. Blocking either Mpro or PLpro prevents the progression of the viral replication cycle, and therefore both proteases represent promising targets for the development of antiviral compounds[8,9]. Importantly, both proteases are highly conserved across coronaviruses, making them potential targets for broad-spectrum antivirals. Viral proteases are known to be suitable targets to prevent virus replication, as several small-molecule inhibitors of HIV and HCV proteases are already in clinical use to treat patients suffering from AIDS or hepatitis C, respectively[10,11].

Current efforts to identify coronavirus protease inhibitors rely on suboptimal approaches. For instance, structure-based in silico screenings or in vitro assays with purified recombinant proteases[12,13] do not take into account the cell permeability or cytotoxicity of prospective compounds. Furthermore, biochemical studies are performed in artificial conditions lacking the complexity of the cellular environment. Alternatively, methods based on live virus replication in cell culture require BSL-3 containment, which is not suitable for handling large sets of compounds, and it is often difficult to discriminate between the antiviral activity and cytotoxicity of compounds, as cytotoxic compounds also impact virus replication. Some Mpro and PLpro cell-based assays compatible with BSL-2 or BSL-1 have recently been proposed[14–16], but they lack robustness, suffering either from a narrow dynamic range or a low signal-to-noise ratio, or they work by measuring reporter signal loss when protease activity is inhibited, a strategy that is also prone to false-positive results in the presence of cytotoxic compounds[17–19].

In addition to their value as targets for antiviral drug discovery, viral proteases represent powerful tools in synthetic biology to control cellular functions. The tobacco etch virus protease (TEVp) has been one of the most extensively explored proteases to transduce input signals into user-defined outputs[20,21]. For instance, split TEV moieties have been fused to conditional dimerization domains, such that protease activity is reconstituted in the presence of the dimerizing ligand, thereby cleaving a target to activate downstream events[22], or to transduce receptor signaling upon ligand binding by releasing transcription factors that can then translocate to the nucleus to regulate gene expression[20,23–25]. However, these tools can only control proteases indirectly. To gain direct control over a protease we need specific protease inhibitors, which can then be used to control

cellular functions, such as modulating the stability of target proteins by inhibiting the cleavage of attached degrons[26].

In this work, we capitalize on synthetic biology tools to develop Mpro and PLpro protease-sensitive gene switches that can accurately report protease activity/inhibition by increasing the expression of a reporter gene in the presence of a specific inhibitor. We design synthetic transcription factors whose transactivation ability is dependent upon protease inhibition and applied them to profile cleavage site preference, to measure the effect of mutations on the protease, and to report on the intracellular inhibitory activity of compounds. Orthogonal Mpro and PLpro-sensitive gene switches were multiplexed to allow potent protease inhibitors to regulate the expression of two transgenes simultaneously. Furthermore, we apply them to program Boolean expression logic controlled by two inputs. We demonstrate the utility of these gene switches to report on the activity of prospective protease inhibitors in vivo, as a proxy for initial bioavailability and pharmacokinetic studies, avoiding the need for BSL-3 virus infection mouse experiments, and we show that these switches can be used to control transgene expression by employing coronavirus protease inhibitors as inputs. The robustness of these coronavirus protease-sensitive gene switches, which we call modular tunable autoproteolytic gene switches (TAGS), makes them promising tools to accelerate the drug development process and to provide precise control of next-generation cell- and gene-based therapies with minimal off-target effects.

## Results

**Design of coronavirus protease-sensitive gene switches**. We set out to create synthetic transcription factors (synTF) able to report on the activity of coronavirus proteases by activating reporter gene expression dose-dependently in response to protease inhibitors. Modular DNA binding (DB) and transcriptional activation (TA) domains are the core components of synTF. We fused them via a protease cleavage site (CS) and attached the corresponding protease to the N-terminus of the DB domain with a flexible GS linker (Fig. 1a). In this configuration, all components are expressed as one polyprotein, which self-inactivates its transactivation ability by proteolytically cleaving off the TA domain. Self-inactivation is prevented in the presence of protease inhibitors that block the proteolytic activity, preserving the TA domain attached to the DB domain. The latter can then bind to specific DNA sequences in promoter regions, thereby activating the expression of downstream transgenes.

Initially, we built SARS-CoV-2 protease activity sensors by fusing the tetracycline repressor TetR (DB domain) with VP16 (TA domain) using linkers containing either PLpro or Mpro cleavage sites and placed human codon-optimized sequences of SARS-CoV-2 PLpro and Mpro proteases in the N-terminus of TetR (synTF$_{PLpro}$ and synTF$_{Mpro}$). The cleavage efficiency determines the performance of the gene switches. Therefore, in order to achieve the highest signal-to-noise ratio, we sought to find the most efficiently cleaved sequences from all the native CSs within the SARS-CoV-2 polyproteins (three $_{PLpro}CS_{NAT(1-3)}$ (Supplementary Table 1) and eleven $_{Mpro}CS_{NAT(1-11)}$) plus an optimized CS ($_{Mpro}CS_{OPT}$) previously identified as being cleaved at higher rates by SARS-CoV Mpro[27] (Supplementary Table 2). Functional synTF can bind to tet operator (tetO) DNA binding sites upstream of a minimal promoter and activate reporter nanoLuc luciferase expression (pNF151, O$_{tet}$-P$_{hCMVmin}$_ss-nano-Luc-sTRSV-pA). We transfected HEK293T cells with the reporter plasmid together with each protease-sensitive synTF and obtained different secretion rates of nanoLuc for synTF bearing different CSs (Fig. 1b, c). From the synTF containing the three native

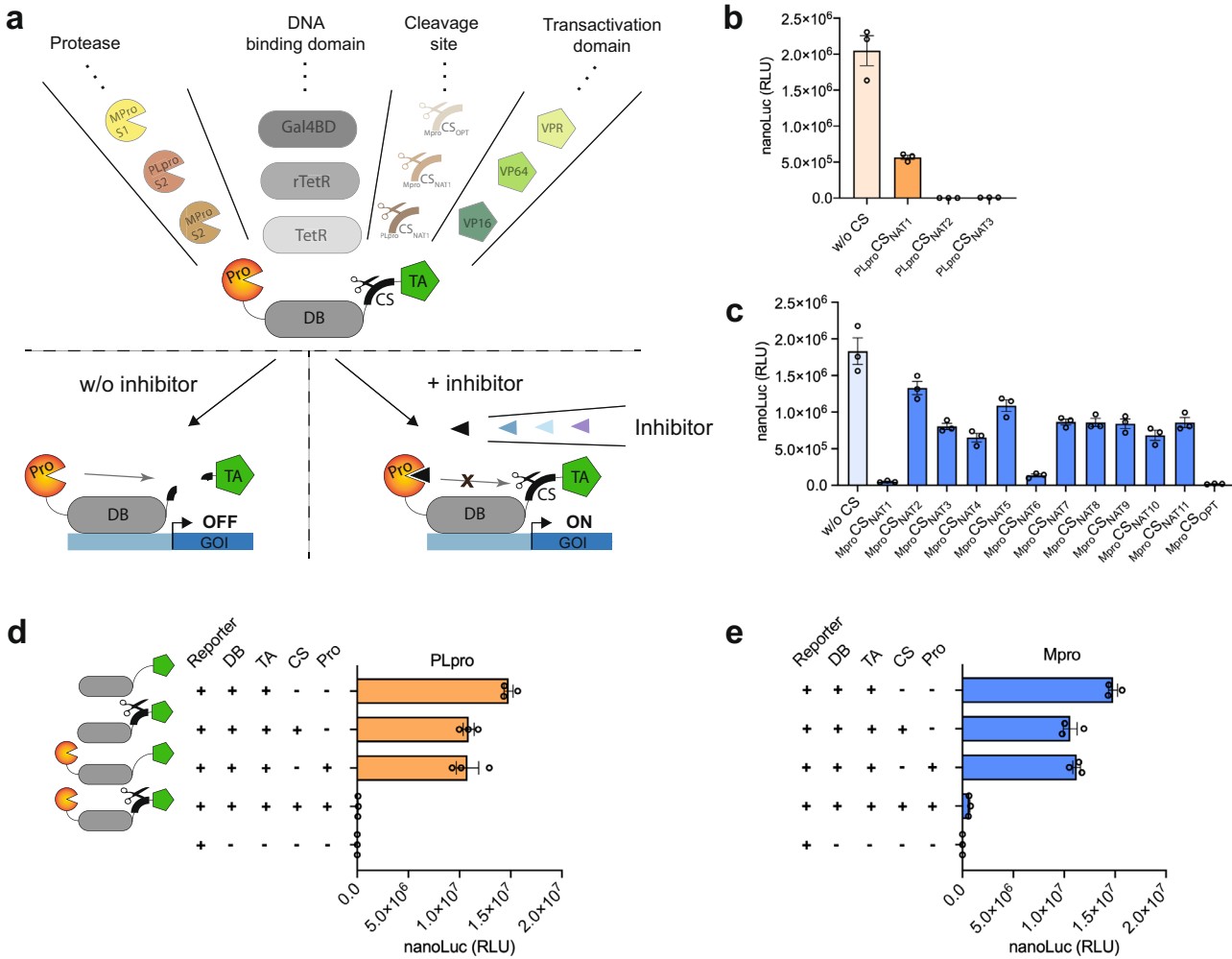

**Fig. 1 Design and optimization of tunable autoproteolytic gene switches (TAGS). a** Schematic of TAGS. The modularity of TAGS enables easy exchange of individual components in a plug-and-play manner, selecting from a range of different proteases, DNA binding (DB) domains, cleavage sites (CS), and transactivation (TA) domains to obtain the desired synthetic transcription factor (synTF) and test inducibility by protease inhibitors. In the absence of a protease inhibitor, synTF self-inactivates through proteolytic cleavage of the TA domain, turning transgene expression OFF. A protease inhibitor blocks the autoproteolysis of synTF and turns transgene expression ON. **b** Scoring PLpro cleavage site preference. nanoLuc luminescence when HEK293T cells were transfected with PLpro-based TAGS bearing one of the three native PLpro CSs from the SARS-CoV-2 polyprotein. PLpro-TAGS without CS was used as a control. **c** Scoring Mpro cleavage site preference. nanoLuc luminescence when HEK293T cells were transfected with Mpro-based TAGS bearing one of the eleven native Mpro CSs from the SARS-CoV-2 polyprotein or an optimized CS. Mpro-TAGS without CS was used as a control. **d** Changes in tTA (TetR-VP16) transactivation ability when fused to PLpro and/or when having a PLpro CS between TetR and VP16. Luminescence from nanoLuc after co-transfecting HEK293T cells with a tetracycline-responsive promoter driving nanoLuc expression (pNF151) and the indicated constructs: non-modified synTF (TetR-VP16), synTF containing CS (TetR-$_{PLpro}$CS$_{NAT3}$-VP16), synTF containing PLpro (PLpro(S2)-TetR-VP16), and synTF containing both PLpro and CS (PLpro(S2)-TetR-$_{PLpro}$CS$_{NAT3}$-VP16). **e** Changes in tTA transactivation ability when fused to Mpro and/or when having a Mpro CS between TetR and VP16. Luminescence from nanoLuc after co-transfecting HEK293T cells with pNF151 and the indicated constructs: non-modified synTF (TetR-VP16), synTF containing CS (TetR-$_{Mpro}$CS$_{OPT}$-VP16), synTF containing Mpro (Mpro(S2)-TetR-VP16), and synTF containing both Mpro and CS (Mpro(S2)-TetR-$_{Mpro}$CS$_{OPT}$-VP16). Data are shown as mean ± SEM, with individual data points, $n = 3$ biological replicates. Source data for this figure is provided in a Source Data file.

PLpro CSs, the $_{PLpro}$CS$_{NAT1}$ based synTF showed the highest reporter expression level with only fourfold less secreted nanoLuc than the synTF lacking any CS, while $_{PLpro}$CS$_{NAT2}$ and $_{PLpro}$CS$_{NAT3}$ were more efficiently cleaved, producing up to 1320-fold less nanoLuc than the synTF without CS (Fig. 1b). The Mpro-sensitive synTF also displayed different cleavage efficiencies, with $_{Mpro}$CS$_{NAT1}$ (sequence between the nsp4 and nsp5) providing the highest cleavage rate and $_{Mpro}$CS$_{NAT2}$ (sequence between nsp5 and nsp6) the lowest among the native CSs (Fig. 1c). The synTF containing $_{Mpro}$CS$_{OPT}$ showed the lowest expression level of nanoLuc (93-fold less than synTF lacking any

CS), suggesting that this is the sequence most efficiently cleaved by Mpro. Profiling protease CSs is widely used to determine amino acid preferences at each CS position, giving insights that serve as a basis to design protease inhibitors.

We performed a set of control experiments to further assess how TetR-VP16 activity is affected by introducing the modifications required to work as a protease activity sensor. We tested synTF modified only with the cleavage sites (pNF152, P$_{PGK}$_TetR-$_{PLpro}$CS$_{NAT3}$-VP16; pNF138, P$_{PGK}$_TetR-$_{Mpro}$CS$_{OPT}$-VP16) or only with the proteases (pNF148, P$_{PGK}$_PLpro(S2)-TetR-VP16; pNF144, P$_{PGK}$_Mpro(S2)-TetR-VP16). Both changes

robustly induced similar nanoLuc expression levels, which were only slightly lower than those achieved with the unmodified non-cleavable TetR-VP16 (Fig. 1d, e). Therefore, none of the modifications have a detrimental effect on synTF activity. Only when both the protease and its target CS are present in the synTF is nanoLuc expression severely hampered, supporting the idea that the proteases retain their activity and successfully cleave the TA domain VP16, leading to synTF inactivation. We also evaluated the cleavage efficiency when the proteases are not fused to synTF, but are provided in trans by expression from separate plasmids. Co-transfection of HEK293T with PLpro (pNF142, $P_{PGK}$_PLpro(S2)-pA) and synTF containing either PLpro CS ($P_{PGK}$_TetR-$_{PLpro}CS_{NAT3}$-VP16) or Mpro CS ($P_{PGK}$_TetR-$_{Mpro}CS_{OPT}$-VP16) did not result in significant differences in nanoLuc expression (Supplementary Fig. 1a). Moreover, co-expressing Mpro (pNF140, $P_{PGK}$_Mpro(S2)-pA) with synTF containing either Mpro CS ($P_{PGK}$_TetR-$_{Mpro}CS_{OPT}$-VP16) or PLpro CS ($P_{PGK}$_TetR-$_{PLpro}CS_{NAT3}$-VP16) only produced a twofold difference in nanoLuc expression levels (Supplementary Fig. 1b). Therefore, the constructs providing the proteases in cis are more suitable for the design of protease inhibitor-inducible switches.

**PLpro inhibitor can regulate transgene expression**. To test whether the PLpro-based gene switch can be induced by PLpro inhibitors, we selected the compound GRL-0617, which is a potent, selective, and competitive noncovalent inhibitor of PLpro[28] and has been shown to block SARS-CoV-2 replication in cell cultures[8]. Exposing synTF$_{PLpro}$-expressing cells to GRL-0617 at low micromolar concentrations triggered reporter gene expression, reaching 40-fold induction over non-treated cells at the highest concentration tested (Fig. 2a). When using YPet as a fluorescent reporter (pNF196, $O_{tet}$-$P_{hCMVmin}$-YPet-sTRSV-pA), we also observed increased fluorescence in the presence of GRL-0617 (Fig. 2b). The performance of PLpro-based tunable autoproteolytic gene switches (PLpro-TAGS) was greatly improved by adding a hammerhead ribozyme (sTRSV) to the 3′UTR of the reporter gene; this is known to decrease the mRNA half-life by cleaving off the polyA[29]. This modification increased both the dynamic range and the GRL-0617-based inducibility of the system, greatly decreasing the leaky expression of nanoLuc in untreated cultures (Supplementary Fig. 2). Next, we characterized the kinetics and reversibility of transcriptional activation by PLpro-TAGS. The results show reversible switching of nanoLuc expression when GRL-0617-containing medium was changed to GRL-0617-free medium, and vice versa (Fig. 2c), as well as fast transactivation kinetics, with significantly increased nanoLuc expression over non-induced cultures at 4 h after GRL-0617 exposure (Fig. 2d). Furthermore, performance profiling in other human cell lines confirmed inducibility by GRL-0617, suggesting the broad applicability of the PLpro gene switch (Fig. 2e). Finally, to test its modularity and show that it can be easily adapted to study PLpro variants from future circulating coronaviruses, we replaced SARS-CoV-2 with SARS-CoV PLpro in the synTF$_{PLpro}$ (pNF180, $P_{PGK}$_PLpro(S1)-TetR-$_{PLpro}CS_{NAT3}$-VP16-pA). These proteases share 83% identity[8]. A preliminary experiment showed that SARS-CoV-derived PLpro successfully cleaves the same substrate as SARS-CoV-2 PLpro (Supplementary Fig. 3) Expression of nanoLuc was highly activated in cells co-transfected with synTF$_{PLpro-S1}$ and pNF151 upon treatment with different concentrations of GRL-0617, reaching up to 120-fold activation over non-induced cultures (Fig. 2f). Head-to-head comparison of PLpro-TAGS with the previously reported FlipGFP-based sensor[14,16] established the superiority of PLpro-TAGS under the tested conditions (Supplementary Fig. 4). Dose-response

experiments using the GRL-0617 inhibitor established that PLpro-TAGS is (i) more sensitive, as it can be triggered at lower concentrations and (ii) has higher inducibility compared to the FlipGFP-PLpro assay. Furthermore, the FlipGFP-PLpro assay takes longer to perform, since cells are treated for 48 h with the test compounds.

**Mpro inhibitor can regulate transgene expression**. To test the inducibility of the Mpro-based gene switch we selected the small-molecule GC-376, as it has been shown to inhibit SARS-CoV Mpro by covalently binding to Cys145 in the Mpro active site and to block SARS-CoV-2 replication in vitro in cultured cells[13]. HEK293T cells co-transfected with synTF$_{Mpro}$ and pNF151, treated with GC-376 showed activation of nanoLuc expression versus non-treated cells at GC-376 concentrations as low as 1 μM, with a maximal 19-fold induction when cells were exposed to 200 μM GC-376 (Fig. 3a). We also compared the performance of the GC-376-induced Mpro sensor containing three different transactivation domains, VP16, VP64, and VPR (Fig. 3b). VPR showed the highest basal expression and therefore the lowest fold induction, while VP64 performed slightly better than the VP16-based sensor and was thus selected as a TA for synTF$_{Mpro}$. As for the PLpro activity sensor, kinetics experiment showed significantly increased reporter gene expression at 4 h after exposure to GC-376 versus non-exposed cells (Fig. 3c). Finally, we tested whether the Mpro sensor could also be adapted for the Mpro from the two previous deadly coronaviruses, SARS-CoV and MERS, which share 96%[30] and 50%[31] sequence identity, respectively, with SARS-CoV-2 Mpro. Preliminary experiments showed that SARS-CoV and MERS-derived Mpro cleave the same substrate as SARS-CoV-2 Mpro (Supplementary Fig. 5a, b). When both sensors were challenged with GC-376, we observed increased reporter nanoLuc expression in a dose-dependent manner (Fig. 3d). The MERS Mpro sensor showed higher basal expression, suggesting that it does not cleave $CS_{OPT}$ as efficiently as the two SARS Mpro variants. In addition, we showed that TAGS can be adapted for proteases from sources beyond the *Coronaviridae* family viruses, exemplified by hepatitis C virus protease (HCVp) (Supplementary Fig. 6).

**Profiling Mpro inhibitors and mutations**. The Mpro-based TAGS was further used to score the potency of reported Mpro inhibitors and to examine the impact of some mutations on Mpro catalytic activity. Other compounds besides GC-376 have been shown to inhibit Mpro, including PF-00835231[32], Mpro-N3[9], 11b[33], GRL-0496[34], and Boceprevir[13], and we assessed their ability to suppress the proteolysis of synTF$_{Mpro}$, thereby activating nanoLuc expression. Of note, the compound PF-00835231 has recently entered phase I clinical trials sponsored by Pfizer[35]. All compounds increased nanoLuc expression over non-induced cultures, although to different extents (Fig. 4a). GRL-0496 and Boceprevir only slightly induced gene expression, whereas PF-00835231, Mpro-N3, and 11b produced a strong, dose-dependent activation of nanoLuc expression, though the response decreased at the highest concentration of 11b (200 μM) due to toxicity. When we tested additional compounds reported to inhibit Mpro in biochemical assays, we found that Mpro-TAGS was not activated by false-positive hits that were ineffective in live-virus assay[36] or by nonspecific inhibitors[37] (Supplementary Fig. 7).

Studying how point mutations affect protease activity is important to identify amino acid residues that are crucial for activity, as this provides knowledge on potential residue clusters that can be targeted for drug development. Thus, we applied our Mpro-sensitive gene switch to screen the effect of three mutations on Mpro activity, namely C145A, H41A, and W31A. These

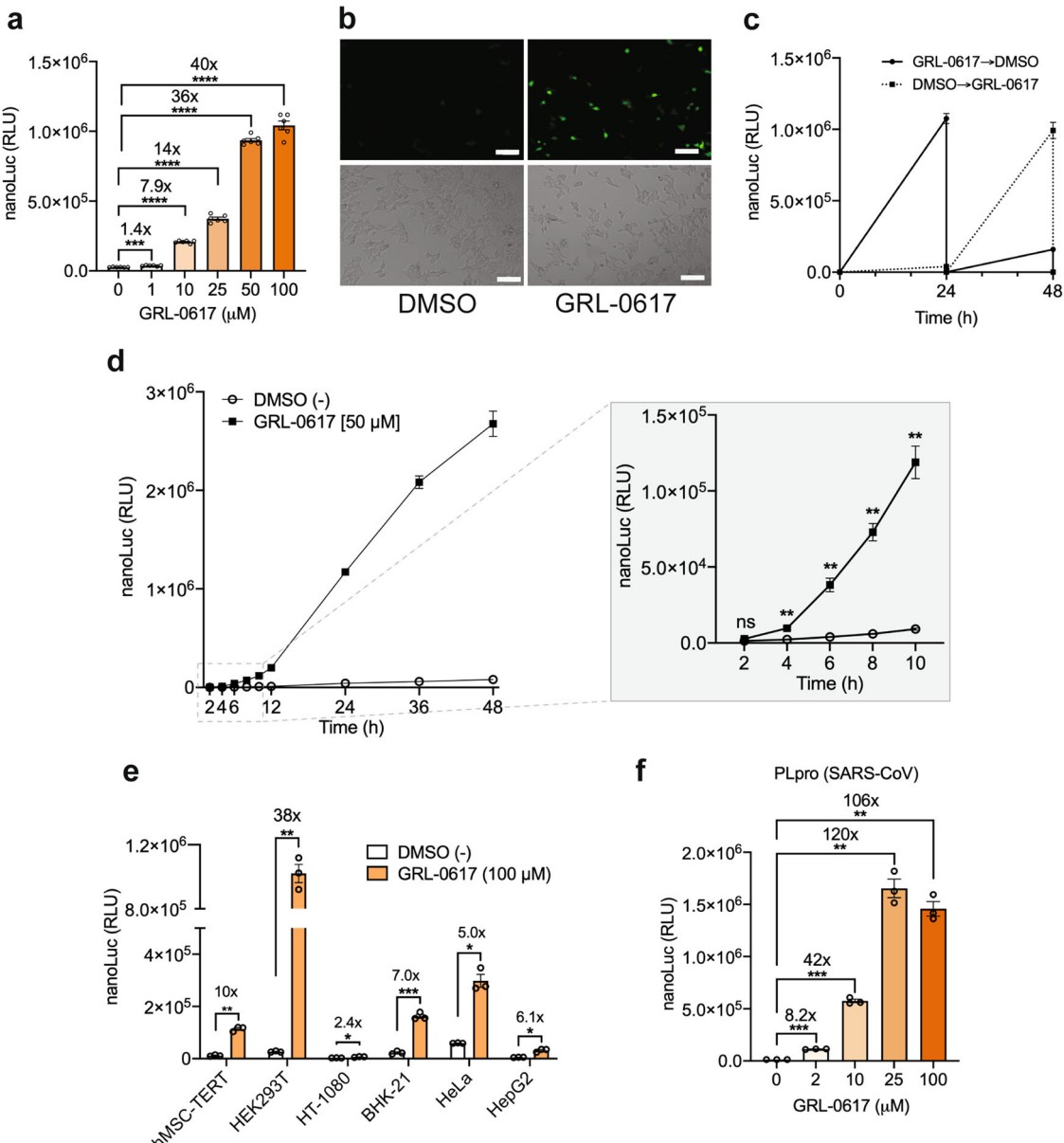

**Fig. 2 PLpro-TAGS characterization. a** Inducibility and sensitivity of PLpro-based TAGS in the presence of GRL-0617. PLpro-TAGS-transfected HEK293T cells (pNF151 and pNF162) were treated with various concentrations of GRL-0617 for 24 h before measuring the nanoLuc expression. **b** Phase-contrast and fluorescence microscopy images of PLpro-TAGS-transfected cells, in which the reporter plasmid encodes the fluorescent YPet protein. Cells were exposed to DMSO or GRL-0617 (50 µM) before microscopy analysis. The scale bar corresponds to 100 µm **c** Reversibility of PLpro-TAGS. Transfected cells were cultured for 48 h, with medium exchange every 24 h, alternating between DMSO and GRL-0617 (50 µM)-containing medium. **d** PLpro-TAGS induction kinetics. PLpro-TAGS-transfected HEK293T cells were grown in a culture medium with or without GRL-0617 for 48 h and luminescence from nanoLuc was measured at various time points. **e** PLpro-TAGS performance in different mammalian cell lines. Mammalian cell lines expressing PLpro-TAGS were treated with DMSO or GRL-0617 before quantifying the nanoLuc expression level. **f** Inducibility and sensitivity of PLpro-TAGS based on SARS-CoV PLpro. Transfected cells were treated with different concentrations of GRL-0617 for 24 h before quantifying nanoLuc expression. Plots **a**, **e**, and **f** show the mean ± SEM, with individual data points, $n = 3$–6 biological replicates. Panel **b** shows representative images from three independent experiments. Plots **c**, **d** show the mean ± SEM, $n = 3$ biological replicates. Statistical significance was calculated by Welch's two-tailed $t$-test, *$P < 0.05$, **$P < 0.01$, ***$P < 0.001$, ****$P < 0.0001$, ns, not significant; exact $P$ values are provided in Supplementary Table 7. Numbers above the bars indicate a fold difference in reporter expression level, calculated by dividing the mean expression level in the presence of inducer by the mean reporter expression level in the absence of inducer. Source data for this figure is provided in a Source Data file.

mutants were generated by site-directed mutagenesis and their activities were compared to that of the wild-type Mpro (Mpro$_{WT}$). C145 and H41 are part of the catalytic site, and both mutations caused elevated expression of nanoLuc relative to Mpro$_{WT}$, suggesting that these mutations greatly impair the catalytic activity of Mpro (Fig. 4b). The W31A mutation, reported

to have a negative impact on 3CL protease from porcine epidemic diarrhea virus[38], had a less severe impact, but still decreased Mpro activity as suggested by a 6.8-fold increase in nanoLuc expression relative to Mpro$_{WT}$. As naturally occurs for all viruses, especially RNA viruses, mutations have emerged in the SARS-CoV-2 genome over time, some of them in the proteases.

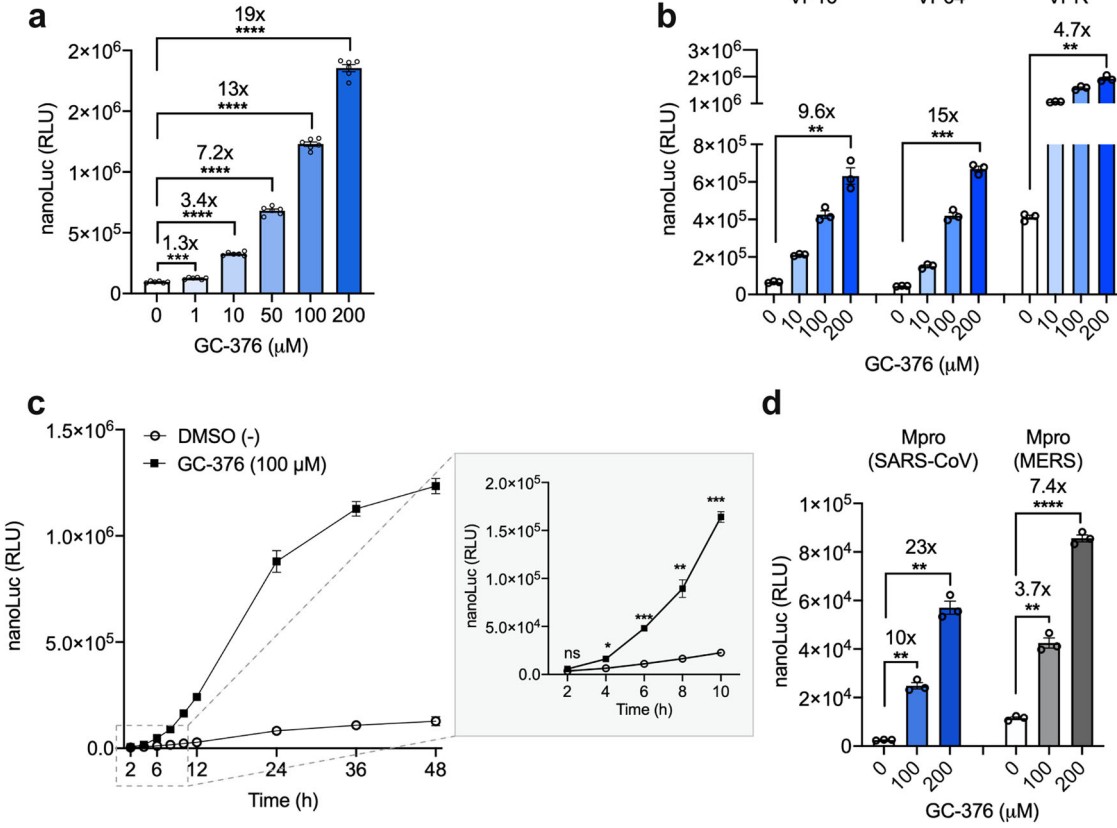

**Fig. 3 Mpro-TAGS characterization. a** Inducibility and sensitivity of the Mpro-TAGS in the presence of GC-376. Mpro-TAGS-transfected HEK293T cells (pNF151 and pNF167) were treated with different concentrations of the Mpro inhibitor GC-376 for 24 h before quantifying nanoLuc expression level. **b** Impact of different transactivation domains (VP16, VP64, and VPR) on Mpro-TAGS performance. HEK293T cells expressing a synTF bearing VP16 (pNF143), VP64 (pNF167), or VPR (pNF157) transactivation domain were treated with various concentrations of GC-376 for 24 h before quantifying nanoLuc expression level. **c** Mpro-TAGS induction kinetics. Mpro-TAGS-transfected HEK293T cells were grown in culture medium with or without GC-376 for 48 h and nanoLuc expression level was quantified at various time points. **d** Inducibility and sensitivity of SARS-CoV and MERS-based Mpro-TAGS. Cells transfected with each Mpro-TAGS were treated with different concentrations of GC-376 for 24 h before quantifying nanoLuc expression level. Plot **a** shows the mean ± SEM, with individual data points, $n = 6$ biological replicates. Plots **b**, **d** show the mean ± SEM, with individual data points, $n = 3$ biological replicates. Plot **c** shows the mean ± SEM, $n = 3$ biological replicates. Statistical significance was calculated by means of Welch's two-tailed $t$-test, *$P < 0.05$, **$P < 0.01$, ***$P < 0.001$, ****$P < 0.0001$, ns, not significant; exact $P$ values are provided in Supplementary Table 7. Numbers above the bars indicate a fold difference in reporter expression level, calculated by dividing the mean expression level in the presence of inducer by the mean reporter expression level in the absence of inducer. Source data for this figure is provided in a Source Data file.

Therefore, we next generated five additional Mpro mutants whose sequences have been found in clinical samples from across the globe[39], namely C156F in the US, G71S in Switzerland, R279C in France, and P184L and a double mutant V77A/K90R in England (Supplementary Table 3). We observed significantly increased nanoLuc expression for all Mpro variants over Mpro$_{WT}$, suggesting that they decrease the activity of Mpro, with P184L causing the largest increase in nanoLuc expression and thus being the most detrimental to Mpro activity (Fig. 4c). A virus with this mutation probably has the least chance of spreading in the population among the variants tested, as in principle a less active Mpro causes the virus to spread at a slower pace. All these enzyme-competent mutants remained sensitive to three selected Mpro inhibitors (Fig. 4d–h). The mutation rate of SARS-CoV-2 is likely to increase to cope with the selection pressure imposed by the administration of vaccines and antivirals. In this context, TAGS will be useful to rapidly study activity differences across emergent protease variants and to screen mutation-resistant inhibitors.

**Boolean logic gates and multiplexing of SARS-CoV-2 protease-sensitive gene switches**. The simple design of the protease

sensors allowed us to build compact two-input logic gates that can compute the presence/absence of protease inhibitors and doxycycline (Dox) into different outputs, accordingly to built-in logical functions (Fig. 5a). Cells expressing each protease sensor (pNF167, P$_{PGK}$_Mpro(S2)-TetR-$_{Mpro}$CS$_{OPT}$-VP64-pA; or pNF162, P$_{PGK}$_PLpro(S2)-TetR-$_{PLpro}$CS$_{NAT3}$-VP16-pA) and treated with combinations of input A (PF-00835231 or GRL-0617) and input B (Dox) showed NIMPLY (A AND NOT B) signal processing (Fig. 5b, c). We also replaced TetR with the reversed tetracycline transactivator (rTetR) (Fig. 5d) in both protease TAGS to obtain AND gate signal processing (Fig. 5e, f).

As the protease-sensitive gene switches are both robustly induced by small molecules, we sought to expand their application to control two target genes simultaneously. For this purpose, we first adapted one of the sensors to work with a different sequence-specific DNA binding domain. Specifically, TetR was replaced by the Gal4 BD domain in the Mpro sensor (pNF186, P$_{PGK}$_Mpro(S2)-GAL4 BD-$_{Mpro}$CS$_{OPT}$-VP64-pA) and co-transfected with its cognate UAS-based reporter plasmid (pAna225, P$_{5xUAS}$_ss-nanoLuc-sTRSV-pA) in HEK293T cells. The transfected cells responded to GC-376 treatment in a

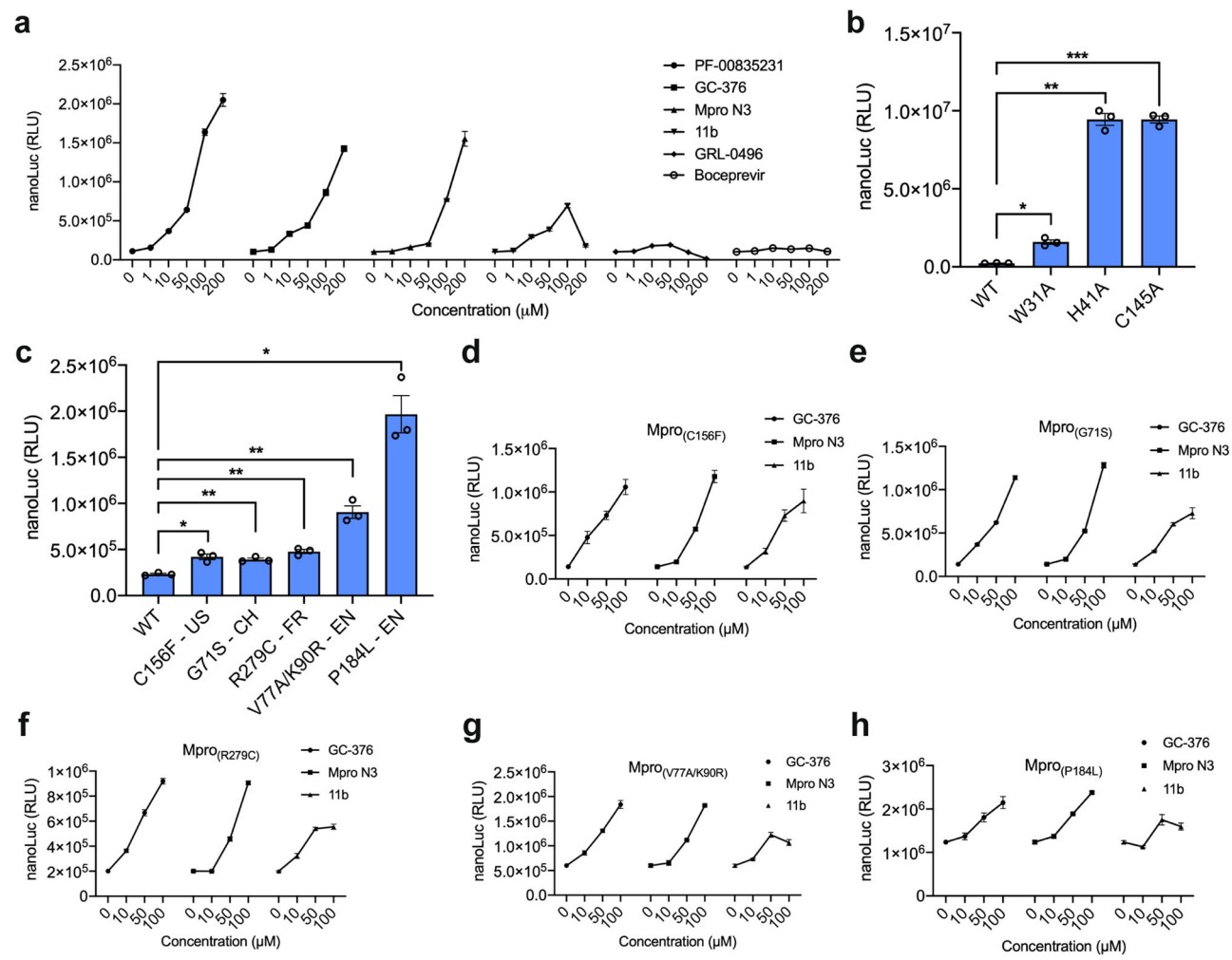

**Fig. 4 Application of Mpro-TAGS for scoring inhibitors and mutations. a** Mpro activity in the presence of various inhibitors. Mpro-TAGS transfected HEK293T cells were treated with various concentrations of selected Mpro inhibitors for 24 h before quantifying nanoLuc expression level. **b** Impact of targeted mutations on Mpro activity. To assess the effect of selected mutations on Mpro activity, nanoLuc levels from HEK293T cells expressing synTF bearing wild-type Mpro or each Mpro mutant (W31A, H41A, and C145A) were compared 36 h post transfection. **c** Activity of Mpro mutants found in clinical samples. NanoLuc secretion from HEK293T cells encoding wild-type or mutant Mpro-TAGS, 36 h after transfection. Mpro mutants were from the USA (C156F), Switzerland (G71S), France (R279C), and England (V77A/K90R and P184L). **d–h** Inhibitor sensitivity of five clinically relevant Mpro mutants. HEK293T cells expressing **d** Mpro(C156F), **e** Mpro(G71S), **f** Mpro(R279C), **g** Mpro(V77A/K90R), and **h** Mpro(P184L)-based TAGS were treated with different doses of GC-376, Mpro-N3 and 11b for 24 h before measuring nanoLuc expression level. Plot **a** shows the mean ± SEM, $n = 3$ biological replicates. Plots **b**, **c** show the mean ± SEM, with individual data points, $n = 3$ biological replicates. Plots **d–h** show the mean ± SEM, $n = 3$ biological replicates. Statistical significance was calculated by means of Welch's two-tailed $t$-test, $*P < 0.05$, $**P < 0.01$, $***P < 0.001$; exact $P$ values are provided in Supplementary Table 7. Source data for this figure is provided in a Source Data file.

dose-dependent manner, reaching up to 45-fold induction of nanoLuc expression (Fig. 5g).

Next, we examined the orthogonality of the two protease sensors, by testing if they could selectively sense their specific inhibitors and produce two different reporter proteins. HEK293T cells were co-transfected with both sensors: (i) the PLpro sensor containing TetR-mediated SEAP expression (pNF162 and pDA326) and (ii) the Mpro sensor containing Gal4-mediated nanoLuc expression (pNF186 and pAna225). Treatment with 50 µM of each inhibitor, either individually or in combination showed selective induction with no crosstalk (Fig. 5h). Only cells treated with both inhibitors showed increased nanoLuc as well as SEAP reporter expression. Treatment with lower doses of inhibitors showed the same pattern of expression, indicating that the orthogonality is maintained across various activation levels (Supplementary Fig. 8). Therefore, the two gene

switches can independently report on each protease activity, retaining orthogonal input-specific gene regulation. Cells engineered with both gene switches could provide a useful cell-based assay for screening large compound libraries to identify specific PLpro and Mpro inhibitors simultaneously, with the advantage that false positives would be easily recognized by activation of both reporter genes. To confirm high-throughput screening compatibility, we treated Mpro-sensing cells seeded in 96-well and 384-well plates with varying concentrations of GC-376 and obtained similar nanoLuc induction levels in both culture formats (Supplementary Fig. 9). Furthermore, we calculated the Z′ factor as a measure of assay robustness[40]. We treated Mpro-TAGS, PLpro-TAGS, or FlipGFP-PLpro transgenic cells growing in 384-well plates either with DMSO (negative control) or 10 µM of the corresponding inhibitor GC-376/GRL-0617 (positive controls). The calculated Z′ factors for Mpro-TAGS and PLpro-TAGS (0.6

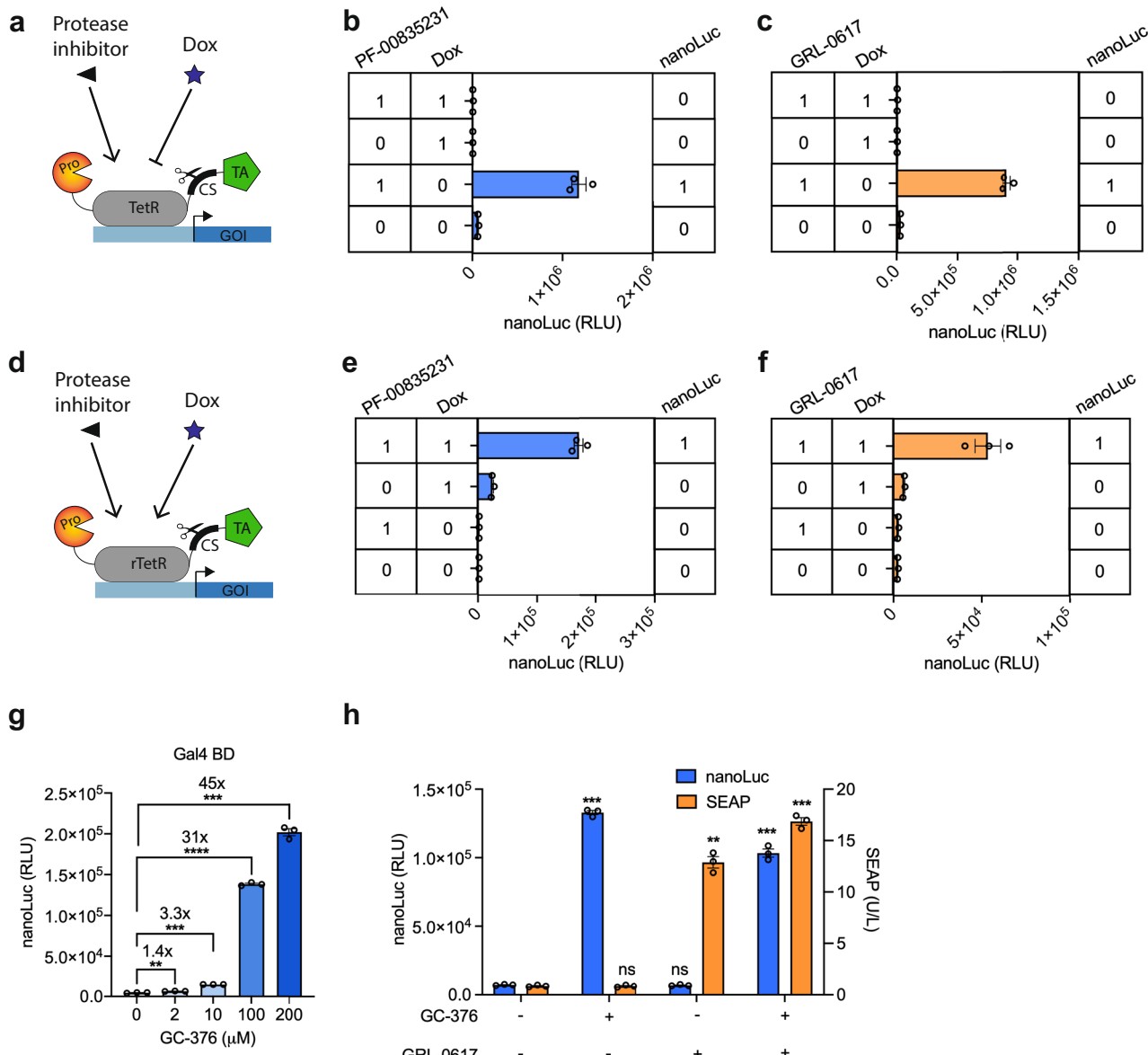

**Fig. 5 Logic operations and multiplexing using TAGS. a** Design of dual-input TAGS able to perform NIMPLY logic function. Transgene expression occurs only in the presence of a protease inhibitor and absence of doxycycline, which can be turned off by adding doxycycline to prevent TetR binding to DNA operator sequence upstream of the transgene. **b, c** Performance of NIMPLY logic gates based on **b** Mpro-TAGS using the Mpro inhibitor PF-00835231 and **c** PLpro-TAGS using PLpro inhibitor GRL-0617. **d** Design of dual-input TAGS able to perform AND logic function. Transgene expression occurs only in the presence of both protease inhibitor and doxycycline which preserve synTF and enable the rTetR to bind to the promoter region. **e, f** Performance of AND logic gates based on **e** Mpro-TAGS using the Mpro inhibitor PF-00835231 and **f** PLpro-TAGS using PLpro inhibitor GRL-0617. **g** Performance of Mpro-TAGS based on Gal4 BD. HEK293T cells co-transfected with pAna225 (P_{5xUAS}_ssnanoLuc-sTRSV-pA) and pNF186 (P_{PGK}_Mpro-Gal4 BD-CS_{OPT}-VP64-pA) were treated with different concentrations of the Mpro inhibitor GC-376 for 24 h before measuring the luminescence of nanoLuc. **h** Multiplexing Mpro- and PLpro-based TAGS. Mpro-TAGS controlling nanoLuc expression and PLpro-TAGS controlling SEAP expression were co-transfected into HEK293T cells, which were then treated with combinations of GC-376 and GRL-0617 at 50 μM for 24 h before profiling reporter expression level. Data are shown as mean ± SEM, with individual data points, $n = 3$ biological replicates. Statistical significance was calculated by means of Welch's two-tailed $t$-test, **$P < 0.01$, ***$P < 0.001$, ****$P < 0.0001$, ns, not significant; exact $P$ values are provided in Supplementary Table 7. Numbers above the bars indicate a fold difference in reporter expression level, calculated by dividing the mean expression level in the presence of inducer by the mean reporter expression level in the absence of inducer. Source data for this figure is provided in a Source Data file.

and 0.66) confirmed the robustness of both assays in a high-throughput setting and, in addition, showed that the FlipGFP-PLpro assay was not robust (Supplementary Fig. 10a–c).

**In vivo performance of TAGS.** Finally, we tested whether the TAGS could be used in vivo to control transgene expression and as a tool to study the activity and bioavailability of small-molecule

protease inhibitors (Fig. 6a). PLpro-TAGS-engineered cells encapsulated in poly-L-lysine-alginate beads were intraperitoneally (i.p) implanted in mice, which were then given either the PLpro inhibitor GRL-0617 or vehicle via oral gavage (o.g). Blood sample analysis revealed significantly increased nanoLuc levels in mice that ingested GRL-0617 (Fig. 6b). Similarly, blood samples taken from mice bearing Mpro-TAGS-engineered cells and given the Mpro inhibitor GC-376 via o.g. showed increased reporter

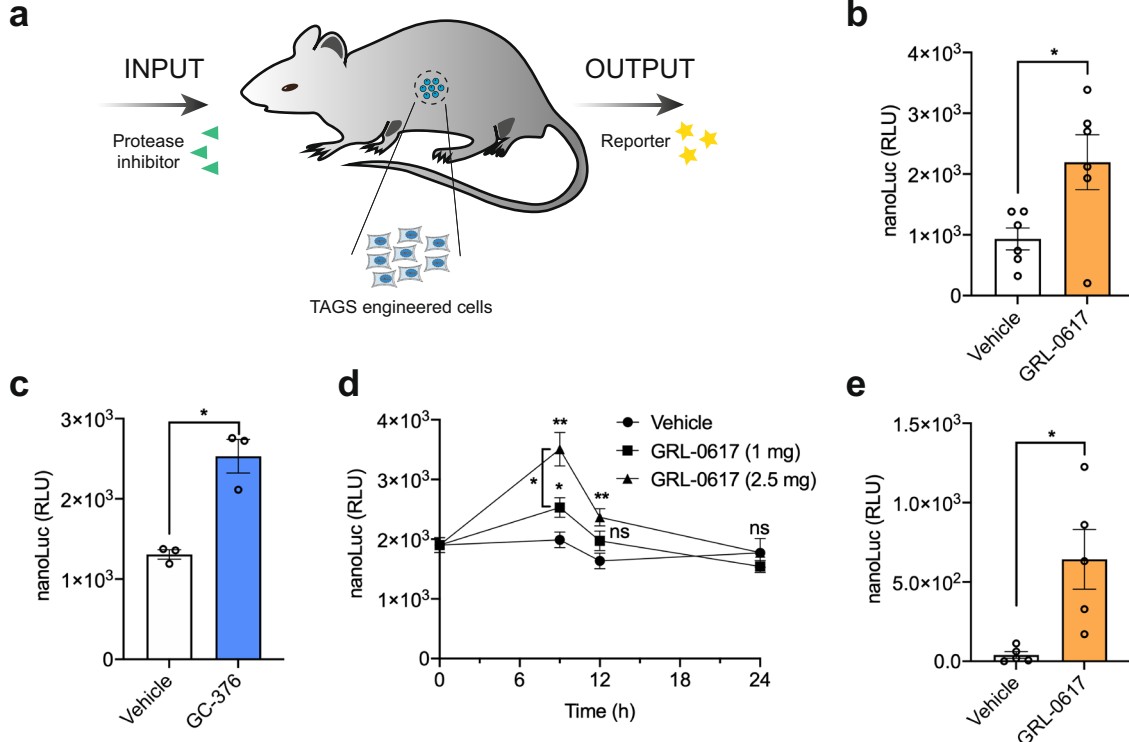

**Fig. 6 TAGS functionality in vivo. a** Schematic of TAGS function in vivo. Protease inhibitor activates encapsulated TAGS-engineered cells or in vivo-transfected cells, resulting in reporter nanoLuc expression. **b** PLpro-TAGS performance in vivo. PLpro-TAGS-engineered cells were intraperitoneally implanted into mice, which then received either GRL-0617 (2.5 mg) or vehicle via o.g.. NanoLuc in the blood was quantified at 6 h thereafter. **c** Mpro-TAGS performance in vivo. Mpro-TAGS-engineered cells were intraperitoneally implanted into mice, which then received either GC-376 (5 mg) or vehicle via o.g.. NanoLuc in the blood was quantified at 6 h thereafter. **d** PLpro-TAGS activation kinetics in vivo. PLpro-TAGS-engineered cells were intraperitoneally implanted into mice, which then received either 1 mg or 2.5 mg of GRL-0617 or vehicle via o.g.. NanoLuc in the blood was quantified at different time points thereafter. **e** Performance of PLpro-TAGS delivered via hydrodynamic injection. PLpro-TAGS-engineered mice received either GRL-0617 (2.5 mg) or vehicle via o.g.. NanoLuc in the blood was quantified at 6 h thereafter. Plot **b** shows the mean ± SEM, with individual data points, $n = 6$ mice. Plot **c** shows the mean ± SEM, with individual data points, $n = 3$ mice. Plot **d** shows the mean ± SEM, $n = 6$ mice. Plot **e** shows the mean ± SEM, with individual data points, $n = 5$ mice. Statistical significance was calculated by means of Welch's two-tailed $t$-test, $*P < 0.05$, $**P < 0.01$, ns, not significant; exact $P$ values are provided in Supplementary Table 7. Source data for this figure is provided in a Source Data file.

nanoLuc expression relative to mice that received only the vehicle (Fig. 6c). These results indicate that both GRL-0617 and GC-376 are sufficiently well absorbed from the gut and remain in the blood for long enough to inhibit the proteases in vivo, thereby activating transgene expression. We also monitored the kinetics of PLpro-TAGS activation in implanted mice given a single low or high dose of the PLpro inhibitor. Analysis of blood samples showed a GRL-0617-dose-dependent increase of nanoLuc levels at 9 h after GRL-0617 ingestion (Fig. 6d). The activation decreased over time, and at 24 h after GRL-0617 administration, treated and non-treated mice had similar basal nanoLuc expression levels, most likely due to clearance of GRL-0617. These results indicate that TAGS could be used to monitor the time course of protease inhibition in vivo. Besides implantation of TAGS-engineered cells, we also tested transfection of the PLpro-TAGS expression plasmids directly into the mice via hydrodynamic injection. PLpro-TAGS-engineered mice that received GRL-0617 via o.g. showed significantly increased nanoLuc levels in blood relative to mice that received only the vehicle (Fig. 6e). These results represent a proof-of-concept that protease-sensitive gene switches can be used in vivo, either as engineered cell implants or via host expression following in vivo transfection, to control the expression of transgenes with small-molecule protease inhibitors. In addition, they can be used as biosensors to report the activity and bioavailability of protease inhibitors during drug development, providing a safer and faster alternative to animal experiments using viruses.

## Discussion

We have designed gene switches based on synthetic autoproteolytic transcription factors that report on the activity of viral proteases and are turned on by inhibitors of those proteases. For the development and validation of these new sense-and-response systems, which we call modular tunable autoproteolytic gene switches (TAGS), we focused on the two proteases of SARS-CoV-2, given the current global threat this virus represents. TAGS enabled us to profile how the protease activity changes in the presence of multiple reported inhibitors, to examine how the activity is affected by several mutations identified in clinical samples, and to score protease cleavage site preference. Thus, TAGS expands the growing set of tools available to control cellular functions by using small-molecule protease inhibitors as input signals[41], and indeed is the first protease-based system that can regulate transgene expression in vivo.

Many Mpro inhibitors so far identified in biochemical assays have not proven to be effective at inhibiting virus replication in cell culture assays[36], and even biochemical assays performed by different labs have yielded contradictory results for the activity of some compounds towards Mpro, raising questions on their validity[42,43]. A likely explanation for the discrepancies is that biochemical assays do not take into account the intracellular native environment of the proteases. In particular, the cell membrane permeability of candidate inhibitors plays an important role in determining their cellular activity. But even if the

compounds can enter the cells, they might undergo enzymatic modifications or bind to nontarget cellular proteins, making them less effective. These characteristics influence the effective concentrations in cells, which might be very different than those found in biochemical assays. The toxicity of a compound is another factor that must be considered, and assays that report a signal increase rather than a decrease in the presence of a candidate inhibitor are preferred in order to avoid false-positive results. Unreliable assays can enormously increase the cost of protease inhibitor discovery if hits are later revealed to be false positives in labor-intensive and biosafety-demanding live virus assays. Although first-generation PLpro and Mpro cell-based assays have been introduced, lack of robustness, low signal-to-noise ratio, and limitation to a single reporter output have restricted their wide adoption.

The TAGS systems described here are not subject to these limitations, as they report on intracellular protease activity and respond to protease inhibitors with a reporter signal increase, thereby decreasing the likelihood of false-positive results. Both PLpro and Mpro-TAGS responded to inhibitor treatment with sensitivities compatible with the effective concentration ranges of inhibitors observed in assays involving live viruses[8,13]. Therefore, TAGS not only accounts for compound toxicity, permeability, and activity but may also give a good estimation of the concentration at which a particular compound is likely to be effective at inhibiting virus propagation in cell culture. PLpro and Mpro-TAGS are both robust, show high inducibility, and have a wide dynamic range. Additionally, the orthogonal design allows the use of multiple different reporters as outputs and even opens up the possibility of multiplex operation to report the activity/inhibition of both PLpro and Mpro simultaneously.

TAGS can be useful beyond SARS-CoV-2, as this modular system would be easily adaptable for other viral proteases. In the last 20 years, we have seen numerous viral outbreaks, three of them being caused by viruses from the *Coronaviridae* family, which suggests that future outbreaks are inevitable and we should be better prepared for them[44]. TAGS is a robust and easy-to-use platform that can be applied for high-throughput screening of compound libraries, and therefore should be effective to speed up antiviral discovery. We have shown here that its individual components (protease, cleavage site, DNA binding domain, and transactivation domain), can be exchanged in a plug-and-play manner. Indeed, we used TAGS to compare the activity of several compounds against PLpro and Mpro from different Coronaviridae family viruses. Mutations can cause protease resistance to inhibitors, as has been observed for HCV and HIV proteases[45,46], and we also showed that TAGS can be quickly adapted to test new clinically relevant protease variants for sensitivity to inhibitors. The incorporation of five Mpro mutants identified in clinical samples into TAGS indicated that they are still enzymatically competent, as expected, given the critical role of Mpro in the virus life cycle, even though they show decreased catalytic activity. TAGS could also be used to test a library of targeted mutants to identify important amino acids involved in the function of the protease, gaining knowledge that would aid the design of small molecules targeting those residues.

Viral protease inhibitors are designed to target nonhuman proteins, making them very promising candidates for chemogenetic regulation of engineered cell-based therapies. Protease inhibitors with proven activity and good pharmacokinetic and pharmacodynamic (PK/PD) properties in preclinical studies, which then show favorable safety and PK/PD profiles during phase I human clinical trials, could be used as inducers for therapeutic delivery by implanted designer cells. The synTF of TAGS has a compact design, and it can be directly controlled by two different inputs to achieve different Boolean logic gates,

providing additional specificity or safety when controlling designer cells. As a proof-of-concept, we used protease inhibitors GRL-0617 and GC-376 to control transgene expression in vivo. We confirmed that designer cells transgenic for PLpro-TAGS or Mpro-TAGS systems, when implanted in mice, show activation of transgene expression in response to ingestion of these compounds by the mice. Furthermore, we used PLpro-TAGS to monitor the kinetics of PLpro inhibition in vivo, showing that TAGS has the potential to be used in drug development studies to assess inhibitor activity in vivo without the need to infect mice with viruses requiring BSL-3 containment. Lastly, we established that hydrodynamic transfection is available to directly engineer mice with TAGS, affording a high fold induction of transgene expression in the presence of protease inhibitors.

Overall, these results suggest that TAGS will be a valuable tool to study viral proteases and to support robust cell-based screening assays for protease inhibitors, as well as to test the antiviral potential of hit compounds for current and future viral threats. We believe this work provides a design foundation for autoproteolytic-based switches that can help to shorten the response time in future viral outbreaks. Furthermore, the compact and modular design of TAGS should make this system particularly useful for the safe and precise control of engineered cell-based therapies.

## Methods

**Plasmid construction**. Design and cloning details for all genetic constructs used in the study are provided in Supplementary Table 4 and Supplementary Table 5. Briefly, constructs were generated by classic molecular biology approaches using restriction enzymes (New England BioLabs) followed by ligation using T4 DNA ligase (ThermoFisher). PCR reactions were performed using Q5 High-Fidelity DNA polymerase (New England BioLabs). All steps were performed according to the manufacturer's instructions. The plasmids were amplified in *Escherichia coli* strain XL10-Gold® (XL10-Gold® ultracompetent cells; 200314, Agilent Technologies). Constructs were verified by Sanger sequencing, done by an external vendor (Microsynth AG). Synthetic gene fragments (Supplementary Table 6) used in the study (e.g., the proteases from SARS-CoV, MERS, and SARS-CoV-2) were first codon-optimized for expression in human cells using Benchling and then synthesized commercially (Twist Bioscience).

**Cell culture and transfection**. Human embryonic kidney cells (HEK293T, ATCC: CRL-11268), hMSC-TERT[47], HT-1080 (ATCC:CCL-121), BHK-21 (ATCC:CCL-10), HeLa (ATCC: CCL-2), and Hep G2 (ATCC: HB-8065) were cultured in Dulbecco's modified Eagle's medium (DMEM; 61965026, ThermoFisher) supplemented with 10% (v/v) fetal bovine serum (FBS; F7524, Sigma-Aldrich) in a humidified atmosphere containing 7.5% $CO_2$ at 37 °C. Cell number and viability were quantified with an electric field multichannel cell device (Casy Cell Counter and Analyzer Model TT; Roche Diagnostics GmbH). For transfection experiments, cells were seeded into 96-well plates (3599, Corning) at $2 \times 10^4$ cells per well, 24 h before transfection. The transfection mixture for one well of 96-well plates consisted of 120 ng of plasmid DNA in 50 μL of FBS-free DMEM and 720 ng polyethyleneimine (PEI, MW 40,000; 24765, Polysciences). After 20 min incubation at room temperature, the transfection mixture was added dropwise to cells, which were then incubated overnight. The next morning, the culture medium was replaced with fresh FBS containing DMEM, and reporter expression was profiled 24 h later. For experiments with inducer treatment, cells were instead transfected in 10 cm cell culture dishes (664160), with transfection mix containing 8000 ng of plasmid DNA and 40 μg PEI mixed in 3.3 mL FBS-free DMEM, seeded the day before at $2 \times 10^6$ cells per plate. The next morning, transfected cells were detached and reseeded at $3.5 \times 10^4$ cells per well into transparent (3599, Corning) or black (655090, Greiner Bio-One) 96-well plate depending on the reporter used, or seeded at $1 \times 10^4$ cells per well into black 384-well plates (3764BC, Corning) and treated with different concentrations of the appropriate inducers or DMSO as a negative control. Reporter expression was profiled 24 h after inducer treatment.

**Inducer preparation**. GC-376 (BG167367, Carbosynth), Mpro-N3 (7230, Tocris), Mpro Inhibitor 11b (31345, Cayman Chemical), GRL-0496 (10-4960-0005, Focus Biomolecules), boceprevir (FB18939, Carbosynth), GRL-0617 (Focus Biomolecules, 10-4965-0005) and PF-00835231 (S9731, Selleckchem), hydroxocobalamin (HY-B2209A, MedChemExpress), Z-DEVD-FMK (HY-12466, MedChemExpress), Z-FA-FMK (S7391, Selleckchem), ebselen (E3520, Sigma), and asunaprevir (HY-14434, MedChemExpress) were prepared as 20 mM solutions in DMSO. Doxycycline hyclate (D9891, Sigma) was prepared as a 50 mg/mL solution in water. Suramin sodium (S2671, Sigma) was prepared as a 20 mM solution in water.

Working solutions were prepared by serially diluting the stock solutions in FBS containing DMEM.

**NanoLuc measurement.** NanoLuc concentrations in cell culture supernatants were quantified with the Nano-Glo Luciferase Assay System (N1110, Promega). About 7.5 μL of cell culture supernatant was mixed with 7.5 μL of buffer/substrate mix (50:1) in 384-well plates (781076, Greiner Bio-One), which were briefly centrifuged at 1200 rpm and incubated for 5 min. Luminescence intensity was measured using a Tecan Infinite M1000 multiplate reader (Tecan AG).

**SEAP measurement.** SEAP levels in cell culture supernatants were quantified by measuring the increase of absorbance due to hydrolysis of para-nitrophenyl phosphate (pNPP). Heat-inactivated (30 min, 65 °C) cell culture supernatant (40 μL) was transferred into a 96-well plate (260836, ThermoFisher) and mixed with 60 μL water, 80 μL 2x SEAP buffer (20 mM homoarginine, 1 mM $MgCl_2$, 21% (v/v) diethanolamine, pH 9.8), and 20 μL substrate solution containing 20 mM pNPP (Acros Organics BVBA). The absorbance of samples was measured at 405 nm using a Tecan Infinite M1000 multiplate reader (Tecan AG).

**Animal experiments.** Cell implants were produced by encapsulating PLpro-TAGS or Mpro-TAGS-transgenic HEK293T cells into coherent alginate-poly-(L-lysine)-alginate beads (400 m; 200 cells/capsule) using an Inotech Encapsulator Research Unit IE-50R (Buechi Labortechnik AG, Flawil, Switzerland) with the following parameters: 25 mL syringe operated at a flow rate of 450 units, 200 μm nozzle with a vibration frequency of 1024 Hz, and bead dispersion voltage of 1.2 kV, stirrer speed set at 4.5 units. Eight-week-old male C57BL6 mice (Charles River Laboratory, Lyon, France) weighing 25–28 g kept at 22 °C, 50% humidity, and 12-h light-dark cycle were intraperitoneally injected with 1 ml serum-free DMEM containing $5 \times 10^6$ cells. GRL-0617, GC-376, or vehicle (10% DMSO, 50% PEG-400, 40% saline) was introduced by oral gavage. Blood samples were collected after treatment and serum was isolated for nanoLuc quantification using BD Microtainer® SST tubes according to the manufacturer's instructions (centrifugation for 5 min at 10 000×g; Becton Dickinson, Plymouth, UK; cat. no. 365967). For hydrodynamic transgene delivery, mice were injected with 2 mL of saline containing plasmid coding for PLpro-TAGS (100 μg DNA per mice) 12 h before starting inhibitor treatment. All experiments involving animals were performed according to the directives of the European Community Council (2010/63/EU), approved by the French Republic (project no. DR2018-40v5 and APAFIS #16753), and carried out by Ghislaine Charpin-El Hamri (no. 69266309) at the University of Lyon, Institut Universitaire de Technologie (IUT), F69622 Villeurbanne Cedex, France.

**Fluorescence imaging and fluorescence intensity measurement.** Single-time-point fluorescence microscopy was performed with a fluorescence microscope (Nikon Eclipse Ti) equipped with a fiber illuminator (Nikon Intensilight C-HGFI), optimized optical filter sets (Semrock), and a digital camera system (Hamamatsu, ORCA Flash 4). The filter set contains a combination of an excitation bandpass filter, emission bandpass filter, and dichroic filter. We measured YPet with 200 ms exposure time and a GFP filter set: HC 470/40, HC 520/35, and BS 495. Images were processed and analyzed using ImageJ software. GFP and mCherry fluorescence intensity of transfected cells was measured in a black 96-well plate (655090, Greiner Bio-One) using a Tecan Infinite M1000 multiplate reader (Tecan AG).

**Reporting Summary.** Further information on research design is available in the Nature Research Reporting Summary linked to this article.

## Data availability

All the data is available in the main text or in the supplementary information materials. Sequence data of plasmids encoding PLpro-TAGS and Mpro-TAGS have been deposited in GenBank under accession codes OK425851, OK425852, and OK425853. Original plasmids are available upon request. All vector information is provided in Supplementary Table 4. Detailed statistical analysis is provided in Supplementary Table 7. Source data is provided in the Source data file. Source data are provided with this paper.

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

## Acknowledgements
This work was supported by the Swiss National Science Foundation National Center of Competence in Research (NCCR) for Molecular Systems Engineering. The authors would like to thank the FAST Lab team for external collaboration from Novartis AG, Ulrich Hassiepen, Laurent Tenaillon and Sandra Lopez Romero for generous advice on cell-based assays and Haijie Zhao for helping with cell encapsulation.

## Author contributions
N.F., A.P.T. and M.F. designed the project, N.F. and A.P.T. conducted in vitro experiments, G.C.-E.H. and S.X. designed and performed the animal experiments, N.F., A.P.T. and M.F. analysed the data and wrote the manuscript.

## Competing interests
The authors declare no competing interests.
