## [Peer Review File · Nature Communications]

Reviewers' Comments:

Reviewer #1:

Remarks to the Author:

The manuscript from Franko et al is an elegant synbio-based approach to screen anti-protease, anti-coronavirus drug candidates. The system is based on a synthetic TF that has cleavage sites for the corona-proteases between the DNA binding domain and the activation domain, fused also to the protease itself. Thus, the protease cleaves the cognate CS, disrupting the function of the TF, unless inhibited by the drug.

The system is simple in the design and with some optimizations the authors obtain a good ON-OFF ratio.

I have just some comments.

Line 139 'to confirm that the protease retain their catalytic activity...'

This line confuses me since the the retainment of their activity is already demonstrated in Fig.1B-C. What shown in Fig 1D-G are controls that could be moved to the supplementary, or combined (at least some of them to the fig. 1B-C

Fig 1H and 1I are missing, as well as Fig.2G

How did the authors make the choice of the Mpro mutations? were they reported already? Also, the entire sequence for the geographical variants tested should be included in the supplementary table.

One last point regarding this paragraph of the manuscript is that I feel it is more relevant to show the response of the mutants to the drugs rather than the catalytic activity itself. Therefore FigS4 should be moved in the Fig 4, and move the fig 4B-c to the supplementary (the latter is a suggestion, but I would definitely move the S4 in the main text).

Reference 22 in line 80 is not pertinent since there is no use of TEVp. The authors should cite instead the CHOMP system by Elowitz lab (DOI: 10.1126/science.aat5062) and the TEVp protein sensing by Siciliano (<https://doi.org/10.1038/s41467-018-03984-5>)

Reviewer #2:

Remarks to the Author:

The authors engineered cell-based systems to screen for the inhibitors of two Sars-CoV-2's proteases. They first examined the design of autoproteolytic gene switches. They then demonstrate the application of the gene switches for the screening of protease inhibitors using HEK293 cells and mice. The main innovation of the work lies in the design of the autoproteolytic gene switches. The switches activate the expression of fluorescence proteins when proteases are inhibited.

The authors have done tremendous work in creating the new cell-based screening systems for Sars-CoV-2 inhibitors. I also commend their contribution to society during this critical time.

However, the work does not seem to significantly advance existing screening systems for inhibitors of Sars-CoV-2's proteases. Furthermore, they have not done enough tests of the screening systems compared to prior work.

1. The work is not the first to create Sars-CoV-2 protease screening systems that fluoresce in the presence of protease inhibitors. As a result, the novelty of the work is low. See one example below:

Development of a Cell-Based Luciferase Complementation Assay for Identification of SARS-CoV-2 3CLpro Inhibitors, *Viruses* 2021, 13(2), 173; <https://doi.org/10.3390/v13020173>

2. Compared to prior work that develops new Sars-CoV-2 protease screening systems, the work

does not test many inhibitors, modify the inhibitors, and validate the results through Sars-CoV-2 infection of relevant cell lines. Without the validation test using Sars-CoV-2, the authors cannot prove that their screening systems can predict the actual efficacy of the inhibitors. As a result, the significance of the new screening systems in the context of the Covid19 pandemic seems low. See one example below:

Discovery of SARS-CoV-2 Papain-like Protease Inhibitors through a Combination of High-Throughput Screening and a FlipGFP-Based Reporter Assay, ACS Cent. Sci. 2021

3. There is potential merit to the authors' suggestion that their reporter is modular and has a lower false-positive rate than other screening systems. However, the claims are not supported by evidence. Modularity should be demonstrated with more proteases. The false-positive rate should be calculated using a large library of inhibitor candidates and be compared to the performance of existing screening assays.

REVIEWER COMMENTS

Reviewer #1 (Remarks to the Author):

The manuscript from Franko et al is an elegant synbio-based approach to screen anti-protease, anti-coronavirus drug candidates. the system is based on a synthetic TF that has cleavage sites for the corona-proteases between the DNA binding domain and the activation domain, fused also to the protease itself. Thus, the protease cleaves the cognate CS, disrupting the function of the TF, unless inhibited by the drug.

The system is simple in the design and with some optimizations the authors obtain a good ON-OFF ratio.

We thank the reviewer for taking her/his time to review the manuscript and give valuable comments to improve it.

I have just some comments.

Line 139 'to confirm that the protease retain their catalytic activity...' This line confuses me since the the retainment of their activity is already demonstrated in Fig.1B-C. What shown in Fig 1D-G are controls that could be moved to the supplementary, or combined (at least some of them to the fig. 1B-C

Indeed, Figs. 1B-C already show that the proteases are catalytically active when fused to the N-terminus of the transcription factor, and we have corrected the text appropriately. We kept Figs. 1 D-E in the main text, since they also show that the function of the transcription factors is not affected by attaching the proteases or inserting the protease cleavage site between the DNA binding and transactivation domains. Following the reviewer's advice, we moved Figs. 1 F-G to the supplementary material, as these panels show an alternative sub-optimal protease activity reporter assay, in which the protease is provided in trans.

Fig 1H and II are missing, as well as Fig.2G

Thank you for pointing this out. These figures were mislabeled in the main text and we have corrected this in the revised version.

How did the authors make the choice of the Mpro mutations? were they reported already?

We selected two mutations involving residues from the SARS-CoV-2 Mpro catalytic site (C145 and H41). In addition, mutation at the W31 residue was reported to impair the activity of the 3CL protease from porcine epidemic diarrhea virus, another member of the Coronaviridae family (ref. 38), so we sought to find out whether it has a similar effect on SARS-CoV-2 Mpro. Various clinically relevant Mpro mutants have been reported (ref. 39), and among those we selected mutants from different geographic locations. We have clarified the reasons for selection of the mutations in the revised manuscript.

Also, the entire sequence for the geographical variants tested should be included in the supplementary table.

We included the amino acid sequences of the Mpro clinical variants in the supplementary material.

One last point regarding this paragraph of the manuscript is that I feel it is more relevant to show the response of the mutants to the drugs rather than the catalytic activity itself. Therefore FigS4 should be moved in the Fig 4, and move the fig 4B-c to the supplementary (the latter is a suggestion, but I would definitely move the S4 in the main text).

We moved Fig. S4 to the main text as suggested by the reviewer. Profiling the catalytic activity of protease mutants can help to identify amino acid residues that are important for protease activity, and that's why we believe it is relevant to show this application of our protease activity reporter system (Figs. 4B-C) in the main text.

Reference 22 in line 80 is not pertinent since there is no use of TEVp. The authors should cite instead the CHOMP system by Elowitz lab (DOI: 10.1126/science.aat5062) and the TEVp protein sensing by Siciliano (<https://doi.org/10.1038/s41467-018-03984-5>)

Thank you. We have replaced the previous reference with the two suggested.

Reviewer #2 (Remarks to the Author):

The authors engineered cell-based systems to screen for the inhibitors of two Sars-CoV-2's proteases. They first examined the design of autoproteolytic gene switches. They then demonstrate the application of the gene switches for the screening of protease inhibitors using HEK293 cells and mice. The main innovation of the work lies in the design of the autoproteolytic gene switches. The switches activate the expression of fluorescence proteins when proteases are inhibited. The authors have done tremendous work in creating the new cell-based screening systems for Sars-CoV-2 inhibitors. I also commend their contribution to society during this critical time.

However, the work does not seem to significantly advance existing screening systems for inhibitors of Sars-CoV-2's proteases. Furthermore, they have not done enough tests of the screening systems compared to prior work.

1. The work is not the first to create Sars-CoV-2 protease screening systems that fluoresce in the presence of protease inhibitors. As a result, the novelty of the work is low. See one example below:

Development of a Cell-Based Luciferase Complementation Assay for Identification of SARS-CoV-2 3CLpro Inhibitors, *Viruses* 2021, 13(2), 173.

We thank the reviewer for taking her/his time to review the manuscript and give valuable comments to improve it. We had already pointed out the advantages of our autoproteolytic gene switches as SARS-CoV-2 protease screening systems over other published systems, and have now also referenced the recently published work identified by the reviewer, which is focused on SARS-CoV-2 Mpro. This system relies on luminescence derived from reconstituted nanoLuc upon Mpro inhibition. The luminescence signal of cells expressing their sensor increased 9.7-fold when treated with 100 μ M GC-376, whereas we obtained a 13-fold luminescence increase with Mpro-TAGS. This higher inducibility and thus better resolution of Mpro-TAGS is important when developing cell-based assays. To further show the potential of TAGS as screening assays, we performed a new experiment in which

we treated Mpro-TAGS transgenic cells growing in 384-well plates either with DMSO (negative control) or 10 μ M GC-376 (positive control). The calculated Z' factor (0.6) confirms the robustness of our assay (new **Fig. S10**).

Another advantage of our autoproteolytic gene switches for antiviral screening is that the modular design allows multiplexing and therefore provides the ability to report on the activity/inhibition of two different proteases in the same system. We have shown TAGS can be combined to report activity/inhibition of both Mpro and PLpro at the same time using two different reporters as outputs. We have expanded on this point in the new version of the manuscript.

2. Compared to prior work that develops new Sars-CoV-2 protease screening systems, the work does not test many inhibitors, modify the inhibitors, and validate the results through Sars-CoV-2 infection of relevant cell lines. Without the validation test using Sars-CoV-2, the authors cannot prove that their screening systems can predict the actual efficacy of the inhibitors. As a result, the significance of the new screening systems in the context of the Covid19 pandemic seems low. See one example below:

Discovery of SARS-CoV-2 Papain-like Protease Inhibitors through a Combination of High-Throughput Screening and a FlipGFP-Based Reporter Assay, ACS Cent. Sci. 2021

The main focus of our work was not to find new inhibitory compounds, but rather to expand the synthetic biology toolbox by providing next-generation tools that could help curb the pandemic we are currently dealing with. Cell-based systems take into account the complex cellular environment, and can thus be very useful to complement biochemically based assays. We have expanded the number of inhibitors tested with Mpro-TAGS (new **Fig. S7**). Some of the compounds were previously validated in relevant cell cultures infected with SARS-CoV-2 (e.g. hydroxocobalamin, Z-FA-FMK). To address point 3 from reviewer 2, we also tested some compounds identified as hits in biochemical assays, but which then failed to inhibit virus replication. It's noteworthy that TAGS did not identify these compounds as hits, in agreement the conclusion that the compounds were false-positive in the original assays.

We appreciate the reference mentioned by the reviewer, and have considered it in the revised version of the manuscript. We conducted a comparative study of the performance of our PLpro-TAGS and the FlipGFP-PLpro reporter assay. Dose-response experiments using the GRL-0617 inhibitor (new **Fig. S4**) showed that PLpro-TAGS is i) more sensitive, as it can be triggered at lower concentrations, and ii) has higher inducibility compared to the FlipGFP-PLpro assay. Furthermore, the FlipGFP-PLpro assay takes longer to perform, since cells are treated for 48 h with the test compounds. The assay duration can be an important consideration when many compounds need to be tested. We have shown that PLpro-TAGS robustly reports protease inhibition 24 h after compound addition and is triggered within 4 h of incubation with the inhibitor, allowing a faster readout of the result. We also compared the Z' factor of both PLpro assays in 384-well plates using GRL-0617 (10 μ M) and DMSO as positive and negative controls, respectively (new **Fig. S10**). The calculated Z' factors indicate that the FlipGFP assay's robustness is quite low, whereas PLpro-TAGS (in the form of Mpro-TAGS) is robust and can be adapted for high-throughput settings.

3. There is potential merit to the authors' suggestion that their reporter is modular and has a lower false-positive rate than other screening systems. However, the claims are not supported by evidence. Modularity should be demonstrated with more proteases. The false-positive rate should be calculated using a large library of inhibitor candidates and be compared to the performance of existing screening assays.

We have demonstrated the modularity of TAGS for five different proteases from the Coronaviridae family. We initially used the two SARS-CoV-2 proteases PL_{pro} and M_{pro} to optimize the system and for proof of principle, and then we validated TAGS for the SARS-CoV PL_{pro} and for M_{pro} from both SARS-CoV and MERS. We have now included a sixth protease, namely the hepatitis C virus protease (HCV_p), to show the application of TAGS outside of the Coronaviridae family. The treatment of HCV_p-TAGS transgenic cells with the inhibitor asunaprevir resulted in high luminescence signal increase in a dose-dependent manner (new **Fig. S6**). Notably, we have demonstrated TAGS modularity at the DNA targeting level by using various DNA-binding domains (TetR and Gal4 DB) and also at the reporter output level (e.g., luminescent protein nanoLuc, fluorescent protein YPet and a phosphatase SEAP for colorimetric assay).

To support the claim that the TAGS systems have a lower false-positive rate than existing screening assays, we tested additional compounds that were identified as hits in biochemical assays using purified protease and cleavable substrate, which is the current gold standard approach to screen protease inhibitors. We tested the 4 most potent and non-cytotoxic compounds (ref 36) in M_{pro}-TAGS transgenic cells. Three false-positive hits (hydroxocobalamin, Z-DEVD-FMK and suramin sodium), which did not inhibit virus replication in cell culture, also did not trigger M_{pro}-TAGS activation. However, M_{pro}-TAGS was activated by the virus replication inhibitor Z-FA-FMK. We also tested one of the first reported M_{pro} inhibitors (ebselen), which was later found to be promiscuous and non-specific, and we observed no activation of M_{pro}-TAGS (new **Fig. S7**). Altogether, these data indicate that TAGS can eliminate at least some of the false-positive “hits” identified in biochemical assays, thus reducing the number of compounds to be taken forward to more cumbersome live virus inhibition assays in BSL-3.

Reviewers' Comments:

Reviewer #2:

Remarks to the Author:

The authors have addressed all my comments.

REVIEWER COMMENTS

Reviewer #1 (Remarks to the Author):

The manuscript from Franko et al is an elegant synbio-based approach to screen anti-protease, anti-coronavirus drug candidates. the system is based on a synthetic TF that has cleavage sites for the corona-proteases between the DNA binding domain and the activation domain, fused also to the protease itself. Thus, the protease cleaves the cognate CS, disrupting the function of the TF, unless inhibited by the drug.

The system is simple in the design and with some optimizations the authors obtain a good ON-OFF ratio.

I have just some comments.

Line 139 'to confirm that the protease retain their catalytic activity...' This line confuses me since the the retainment of their activity is already demonstrated in Fig.1B-C. What shown in Fig 1D-G are controls that could be moved to the supplementary, or combined (at least some of them to the fig. 1B-C

Fig 1H and 1I are missing, as well as Fig.2G

How did the authors make the choice of the Mpro mutations? were they reported already?

Also, the entire sequence for the geographical variants tested should be included in the supplementary table.

One last point regarding this paragraph of the manuscript is that I feel it is more relevant to show the response of the mutants to the drugs rather than the catalytic activity itself. Therefore FigS4 should be moved in the Fig 4, and move the fig 4B-c to the supplementary (the latter is a suggestion, but I would definitely move the S4 in the main text).

Reference 22 in line 80 is not pertinent since there is no use of TEVp. The authors should cite instead the CHOMP system by Elowitz lab (DOI: 10.1126/science.aat5062) and the TEVp protein sensing by Siciliano (<https://doi.org/10.1038/s41467-018-03984-5>)

Reviewer #2 (Remarks to the Author):

The authors engineered cell-based systems to screen for the inhibitors of two Sars-CoV-2's proteases. They first examined the design of autoproteolytic gene switches. They then demonstrate the application of the gene switches for the screening of protease inhibitors using HEK293 cells and mice. The main innovation of the work lies in the design of the autoproteolytic gene switches. The switches activate the expression of fluorescence proteins when proteases are inhibited. The authors have done tremendous work in creating the new cell-based screening systems for Sars-CoV-2 inhibitors. I also commend their contribution to society during this critical time.

However, the work does not seem to significantly advance existing screening systems for inhibitors of Sars-CoV-2's proteases. Furthermore, they have not done enough tests of the screening systems compared to prior work.

1. The work is not the first to create Sars-CoV-2 protease screening systems that fluoresce in the presence of protease inhibitors. As a result, the novelty of the work is low. See one example below:

Development of a Cell-Based Luciferase Complementation Assay for Identification of SARS-CoV-2 3CLpro Inhibitors, *Viruses* 2021, 13(2), 173.

2. Compared to prior work that develops new Sars-CoV-2 protease screening systems, the work does not test many inhibitors, modify the inhibitors, and validate the results through Sars-CoV-2 infection of relevant cell lines. Without the validation test using Sars-CoV-2, the authors cannot prove that their screening systems can predict the actual efficacy of the inhibitors. As a result, the significance of the new screening systems in the context of the Covid19 pandemic seems low. See one example below:

Discovery of SARS-CoV-2 Papain-like Protease Inhibitors through a Combination of High-Throughput Screening and a FlipGFP-Based Reporter Assay, *ACS Cent. Sci.* 2021

3. There is potential merit to the authors' suggestion that their reporter is modular and has a lower false-positive rate than other screening systems. However, the claims are not supported by evidence. Modularity should be demonstrated with more proteases. The false-positive rate should be calculated using a large library of inhibitor candidates and be compared to the performance of existing screening assays.

REVIEWER COMMENTS

Reviewer #1 (Remarks to the Author):

The manuscript from Franko et al is an elegant synbio-based approach to screen anti-protease, anti-coronavirus drug candidates. The system is based on a synthetic TF that has cleavage sites for the corona-proteases between the DNA binding domain and the activation domain, fused also to the protease itself. Thus, the protease cleaves the cognate CS, disrupting the function of the TF, unless inhibited by the drug.

The system is simple in the design and with some optimizations the authors obtain a good ON-OFF ratio.

We thank the reviewer for taking her/his time to review the manuscript and give valuable comments to improve it.

I have just some comments.

Line 139 'to confirm that the protease retain their catalytic activity...' This line confuses me since the the retainment of their activity is already demonstrated in Fig.1B-C. What shown in Fig 1D-G are controls that could be moved to the supplementary, or combined (at least some of them to the fig. 1B-C

Indeed, Figs. 1B-C already show that the proteases are catalytically active when fused to the N-terminus of the transcription factor, and we have corrected the text appropriately. We kept Figs. 1 D-E in the main text, since they also show that the function of the transcription factors is not affected by attaching the proteases or inserting the protease cleavage site between the DNA binding and transactivation domains. Following the reviewer's advice, we moved Figs. 1 F-G to the supplementary material, as these panels show an alternative sub-optimal protease activity reporter assay, in which the protease is provided in trans.

Fig 1H and 1I are missing, as well as Fig.2G

Thank you for pointing this out. These figures were mislabeled in the main text and we have corrected this in the revised version.

How did the authors make the choice of the Mpro mutations? were they reported already?

We selected two mutations involving residues from the SARS-CoV-2 Mpro catalytic site (C145 and H41). In addition, mutation at the W31 residue was reported to impair the activity of the 3CL protease from porcine epidemic diarrhea virus, another member of the Coronaviridae family (ref. 38), so we sought to find out whether it has a similar effect on SARS-CoV-2 Mpro. Various clinically relevant Mpro mutants have been reported (ref. 39), and among those we selected mutants from different geographic locations. We have clarified the reasons for selection of the mutations in the revised manuscript.

Also, the entire sequence for the geographical variants tested should be included in the supplementary table.

We included the amino acid sequences of the Mpro clinical variants in the supplementary material.

One last point regarding this paragraph of the manuscript is that I feel it is more relevant to show the response of the mutants to the drugs rather than the catalytic activity itself. Therefore FigS4 should be moved in the Fig 4, and move the fig 4B-c to the supplementary (the latter is a suggestion, but I would definitely move the S4 in the main text).

We moved Fig. S4 to the main text as suggested by the reviewer. Profiling the catalytic activity of protease mutants can help to identify amino acid residues that are important for protease activity, and that's why we believe it is relevant to show this application of our protease activity reporter system (Figs. 4B-C) in the main text.

Reference 22 in line 80 is not pertinent since there is no use of TEVp. The authors should cite instead the CHOMP system by Elowitz lab (DOI: 10.1126/science.aat5062) and the TEVp protein sensing by Siciliano (<https://doi.org/10.1038/s41467-018-03984-5>)

Thank you. We have replaced the previous reference with the two suggested.

Reviewer #2 (Remarks to the Author):

The authors engineered cell-based systems to screen for the inhibitors of two Sars-CoV-2's proteases. They first examined the design of autoproteolytic gene switches. They then demonstrate the application of the gene switches for the screening of protease inhibitors using HEK293 cells and mice. The main innovation of the work lies in the design of the autoproteolytic gene switches. The switches activate the expression of fluorescence proteins when proteases are inhibited. The authors have done tremendous work in creating the new cell-based screening systems for Sars-CoV-2 inhibitors. I also commend their contribution to society during this critical time.

However, the work does not seem to significantly advance existing screening systems for inhibitors of Sars-CoV-2's proteases. Furthermore, they have not done enough tests of the screening systems compared to prior work.

1. The work is not the first to create Sars-CoV-2 protease screening systems that fluoresce in the presence of protease inhibitors. As a result, the novelty of the work is low. See one example below:

Development of a Cell-Based Luciferase Complementation Assay for Identification of SARS-CoV-2 3CLpro Inhibitors, *Viruses* 2021, 13(2), 173.

We thank the reviewer for taking her/his time to review the manuscript and give valuable comments to improve it. We had already pointed out the advantages of our autoproteolytic gene switches as SARS-CoV-2 protease screening systems over other published systems, and have now also referenced the recently published work identified by the reviewer, which is focused on SARS-CoV-2 Mpro. This system relies on luminescence derived from reconstituted nanoLuc upon Mpro inhibition. The luminescence signal of cells expressing their sensor increased 9.7-fold when treated with 100 μ M GC-376, whereas we obtained a 13-fold luminescence increase with Mpro-TAGS. This higher inducibility and thus better resolution of Mpro-TAGS is important when developing cell-based assays. To further show the potential of TAGS as screening assays, we performed a new experiment in which we treated Mpro-TAGS transgenic cells growing in 384-well plates either with DMSO (negative control) or 10 μ M GC-376 (positive control). The calculated Z' factor (0.6) confirms the robustness of our assay (new **Fig. S10**).

Another advantage of our autoproteolytic gene switches for antiviral screening is that the modular design allows multiplexing and therefore provides the ability to report on the activity/inhibition of two different proteases in the same system. We have shown TAGS can be combined to report activity/inhibition of both Mpro and PLpro at the same time using two different reporters as outputs. We have expanded on this point in the new version of the manuscript.

2. Compared to prior work that develops new Sars-CoV-2 protease screening systems, the work does not test many inhibitors, modify the inhibitors, and validate the results through Sars-CoV-2 infection of relevant cell lines. Without the validation test using Sars-CoV-2, the authors cannot prove that their screening systems can predict the actual efficacy of the inhibitors. As a result, the significance of the new screening systems in the context of the Covid19 pandemic seems low. See one example below:

Discovery of SARS-CoV-2 Papain-like Protease Inhibitors through a Combination of High-Throughput Screening and a FlipGFP-Based Reporter Assay, *ACS Cent. Sci.* 2021

The main focus of our work was not to find new inhibitory compounds, but rather to expand the synthetic biology toolbox by providing next-generation tools that could help curb the pandemic we are currently dealing with. Cell-based systems take into account the complex cellular environment, and can thus be very useful to complement biochemically based assays. We have expanded the number of inhibitors tested with Mpro-TAGS (new **Fig. S7**). Some of the compounds were previously validated in relevant cell cultures infected with SARS-CoV-2 (e.g. hydroxocobalamin, Z-FA-FMK). To address point 3 from reviewer 2, we also tested some compounds identified as hits in biochemical assays, but which then failed to inhibit virus replication. It's noteworthy that TAGS did not identify these compounds as hits, in agreement the conclusion that the compounds were false-positive in the original assays. We appreciate the reference mentioned by the reviewer, and have considered it in the revised version of the manuscript. We conducted a comparative study of the performance of our PLpro-TAGS and the FlipGFP-PLpro reporter assay. Dose-response experiments using the GRL-0617 inhibitor (new **Fig. S4**) showed that PLpro-TAGS is i) more sensitive, as it can be triggered at lower concentrations, and ii) has higher inducibility compared to the FlipGFP-PLpro assay. Furthermore, the FlipGFP-PLpro assay takes longer to perform, since cells are treated for 48 h with the test compounds. The assay duration can be an important consideration when many compounds need to be tested. We have shown that PLpro-TAGS robustly reports protease inhibition 24 h after compound addition and is triggered within 4 h of incubation with the inhibitor, allowing a faster readout of the result. We also compared the Z' factor of both PLpro assays in 384-well plates using GRL-0617 (10 μ M) and DMSO as positive and negative controls, respectively (new **Fig. S10**). The calculated Z' factors indicate that the FlipGFP assay's robustness is quite low, whereas PLpro-TAGS (in the form of Mpro-TAGS) is robust and can be adapted for high-throughput settings.

3. There is potential merit to the authors' suggestion that their reporter is modular and has a lower false-positive rate than other screening systems. However, the claims are not supported by evidence. Modularity should be demonstrated with more proteases. The false-positive rate should be calculated using a large library of inhibitor candidates and be compared to the performance of existing screening assays.

We have demonstrated the modularity of TAGS for five different proteases from the Coronaviridae family. We initially used the two SARS-CoV-2 proteases PLpro and Mpro to optimize the system and for proof of principle, and then we validated TAGS for the SARS-CoV PLpro and for Mpro from both SARS-CoV and MERS. We have now included a sixth protease, namely the hepatitis C virus protease (HCVp), to show the application of TAGS outside of the Coronaviridae family. The treatment of HCVp-TAGS transgenic cells with the inhibitor asunaprevir resulted in high luminescence signal increase in a dose-dependent manner (new **Fig. S6**). Notably, we have demonstrated TAGS modularity at the DNA targeting level by using various DNA-binding domains (TetR and Gal4 DB) and also at the reporter output level (e.g., luminescent protein nanoLuc, fluorescent protein YPet and a phosphatase SEAP for colorimetric assay).

To support the claim that the TAGS systems have a lower false-positive rate than existing screening assays, we tested additional compounds that were identified as hits in biochemical assays using purified protease and cleavable substrate, which is the current gold standard approach to screen protease inhibitors. We tested the 4 most potent and non-cytotoxic compounds (ref 36) in Mpro-TAGS transgenic cells. Three false-positive hits (hydroxocobalamin, Z-DEVD-FMK and suramin sodium), which did not inhibit virus replication in cell culture, also did not trigger Mpro-TAGS activation. However, Mpro-TAGS was activated by the virus replication inhibitor Z-FA-FMK. We also tested one of the first

reported Mpro inhibitors (ebselen), which was later found to be promiscuous and non-specific, and we observed no activation of Mpro-TAGS (new **Fig. S7**). Altogether, these data indicate that TAGS can eliminate at least some of the false-positive "hits" identified in biochemical assays, thus reducing the number of compounds to be taken forward to more cumbersome live virus inhibition assays in BSL-3.

REVIEWERS' COMMENTS

Reviewer #2 (Remarks to the Author):

The authors have addressed all my comments.

REVIEWERS' COMMENTS

Reviewer #2 (Remarks to the Author):

The authors have addressed all my comments.

We are glad the comments were addressed successfully. Thank you for working with us to improve the manuscript.